# NUISANCE-ROBUST WEIGHTING NETWORK FOR END-TO-END CAUSAL EFFECT ESTIMATION

## ABSTRACT

We combine the two major approaches to causal inference: the conventional statistical approach based on weighting and the end-to-end learning with adversarial networks. Causal inference concerns the expected loss in a distribution different from the training distribution due to intervening on the input variables. Recently, the representation balancing approach with neural networks has repeatedly demonstrated superior performance for complex problems, owing to its end-to-end modeling by adversarial formulation. However, some recent work has shown that the limitation lies in the unrealistic theoretical assumption of the invertibility of the representation extractor. This inherent difficulty stems from the fact that the representation-level discrepancy in representation balancing accounts only for the uncertainty of the later layers than the representation, i.e., the hypothesis layers and the loss. Therefore, we shed light once again on the conventional weighting-based approach, retaining the spirit of end-to-end learning. Most conventional statistical methods are based on inverse probability weighting using propensity scores, which involves nuisance estimation of propensity as an intermediate step. They often suffer from inaccurate estimation of the propensity scores and instability due to large weights. One might be tempted to jointly optimize the nuisance and the target, though it may lead to an optimistic evaluation, e.g., avoiding noisy instances by weighting less when noise levels are heterogeneous. In this paper, we propose a simple method that amalgamates the strengths of both approaches: adversarial joint optimization of the nuisance and the target. Our formulation follows the pessimistic evaluation principle in offline reinforcement learning, which brings provable robustness to the estimation uncertainty of the nuisance and the instability due to extreme weights. Our method performed consistently well under challenging settings with heterogeneous noise. Our code is available online: https://anonymous.4open.science/r/NuNet-002A.

## 1 INTRODUCTION

Causal inference enables us to assess a treatment action's impact on the decision-making process under uncertainty. Its application originated in the policy-making field (LaLonde, 1986) including healthcare (Sanchez et al., 2022). Recently, the focus was expanded to individualized decision-making such as precision medicine (Sanchez et al., 2022), recommendation (Schnabel et al., 2016; Bonner and Vasile, 2018), and advertisement (Sun et al., 2015; Wang et al., 2015) with the help of advanced machine learning-based methods.

We estimate the causal effect of a treatment action (e.g., prescription of a specific medicine). That is, we need accurate predictions of both potential outcomes with and without the treatment to take its difference since the supervision of the actual effect itself is never given. With observational data, where the action is not assigned completely at random but selected by past decision-makers, the model should generalize not only to the factual data distribution but also to the biasedly missing counterfactual outcome of the action that was not selected. This is called the fundamental problem of causal inference (Shalit et al., 2017).

Conventional statistical methods for causal inference deal with this action selection bias by matching extraction or importance sampling at the input feature level (Rosenbaum and Rubin, 1983). Among others, inverse probability weighting using propensity scores (IPW) (Austin, 2011) is a represen-

tative and versatile approach. The IPW method first learns the policy of the past decision makers (propensity) and then trains a target model with weights of the inverse of propensities, i.e., an estimate of the rarity of the observed action. This two-step estimation strategy defines the fundamental limitation of the weighting approach. The problems are twofold: 1) the overall accuracy depends on the precision of the propensity score estimation at the first stage, and 2) even if the propensity score was precise, the weights could be assigned skewed toward a small fraction of the whole training sample, resulting in high estimation variance (Kang et al., 2007). The overall accuracy can only be guaranteed asymptotically, limiting its applicability to modern non-asymptotic situations such as high dimensional models as DNNs for capturing complex heterogeneity or complex action spaces.

Various countermeasures have been tried to alleviate this problem, such as doubly robust (DR) methods (Kang et al., 2007; Kennedy, 2020; Dudík et al., 2014) and double machine learning (Chernozhukov et al., 2018; Nie and Wager, 2021), which are clever combinations of outcome prediction models and only weighting its residuals using estimated propensity. Nevertheless, the IPW-based approach's limitation is the instability of the two-step procedure in non-asymptotic situations. Its large estimation variance is problematic in cases of high dimensional covariates or cases where the propensity score is close to $0$ or $1$ and thus the effective sample size is limited (Athey et al., 2018).

As in various other fields, advancement with deep neural networks (DNN) has gained substantial attention in causal inference literature (Li and Zhu, 2022). One of the notable advancements made when Shalit et al. (2017) and Johansson et al. (2016) introduced DNN to causal inference was representation-level balancing through distribution discrepancy measures. The representation extracted from the input covariates is encouraged to be *balanced*, i.e., independent of the action, by measuring and minimizing the discrepancy between the representation distributions conditioned on the action. While the gold standard of causal inference is randomized controlled trials (RCTs), which are costly in many real-world scenarios, this technique allows the construction of hypotheses on a balanced representation space that would virtually mimic RCTs. The performance of representation balancing is guaranteed by a generalization error upper-bound (Shalit et al., 2017) in a non-asymptotic manner. This results in an end-to-end training procedure, free from the concerns in the intermediate estimation problem.

However, an unrealistic assumption in its theoretical guarantees has been highlighted in several recent studies (Johansson et al., 2019; Zhao et al., 2019; Wu et al., 2020). When the representation becomes degenerate and not invertible, the difference in errors arising from varying input distributions is overlooked. Such oversight stems from the inherent challenge of not addressing uncertainty within the representation extractor The representation discrepancy upper-bounds error only from the uncertainty in the later layers than the representation, i.e., the hypothesis layers and the loss.

**Contribution**  We elevate the conventional two-step weighted estimation with the spirit of end-to-end adversarial training rather than representation-level balancing at risk of bias. We define the worst-case loss with respect to the nuisance model uncertainty inspired by the pessimism principle in offline reinforcement learning. Our adversarial loss simultaneously accounts for the true propensity's uncertainty and the statistical instability due to weighting. We apply our framework to the doubly robust learner (Kennedy, 2020), an extension of IPW. The proposed method performed consistently well on datasets with heterogeneous noise, including real-world datasets, in which non-adversarial losses tend to be too optimistic.

## 2 PROBLEM SETTING

We consider a standard causal inference framework. We have observational data $D = \{(x^{(n)}, a^{(n)}, y^{(n)})\}$ consisting of i.i.d. instances of $d$-dimensional background feature $x^{(n)} \in \mathcal{X} \subset \mathbb{R}^d$, a treatment action $a^{(n)} \in \mathcal{A} = \{0, 1\}$, and a continuous or binary outcome $y^{(n)} \in \mathcal{Y}$, where $n$ is the sample instance index. In the Neyman-Rubin potential outcome framework (Rubin, 2005), the potential outcomes of both factual and counterfactual actions are expressed as random variables $\{Y_a\}_{a \in \{0,1\}}$, of which only the factual outcome is observed $\left(y^{(n)} = y_{a^{(n)}}\right)$ and the counterfactual one $y_{1-a^{(n)}}$ is missing.

Our goal is to learn a potential outcome function $f : \mathcal{X} \times \mathcal{A} \to \mathcal{Y}$ to estimate the causal effect $\hat{\tau}(x) := \hat{f}(x, a = 1) - \hat{f}(x, a = 0)$ under the given background feature $x$, or to learn $\hat{\tau}$ directly. The estimated effect $\hat{\tau}(x)$ is expected to approximate the true individualized causal effect defined as the conditional average treatment effect (CATE).

$$\tau(x) = \mathbb{E}\left[Y_1 - Y_0 | x\right]$$

A typical metric for the estimation accuracy is the MSE of $\tau(x)$, also known as the precision in estimating heterogeneous effects (PEHE).

$$\text{PEHE}(\hat{\tau}) = \mathbb{E}_x\left[(\tau(x) - \hat{\tau}(x))^2\right]$$

As a sufficient condition for consistent learnability of the CATE, we follow the standard set of assumptions in the potential outcome framework (Imbens and Rubin, 2015).

- $Y^{(n)} \perp\!\!\!\perp A^{(n')} \ \forall n \neq n'$ (Stable Unit Treatment Value Assumption)
- $(Y_0, Y_1) \perp\!\!\!\perp A \mid X$ (unconfoundedness)
- $0 < \mu(a|x) < 1 \ \forall x, a$ (overlap)

## 3 RELATED WORK

**Inverse probability weighting with propensity scores (IPW) and its extension**  IPW is a well-known and universal approach to various causal inference problems, where the density is balanced by weighting using estimated propensity scores. The extended approach called orthogonal statistical learning including the DR-Learner (Kennedy, 2020) and the R-Learner (Nie and Wager, 2021) also exploit outcome prediction models. These methods have been shown to be robust to estimation errors for the first-stage nuisance parameter, i.e., the errors do not affect the final estimation in the first-order sense of the Taylor expansion. Nonetheless, high estimation variance is a drawback of this approach when applied to non-asymptotic situations (Athey et al., 2018). Aiming at robustness for complex DNN-based models, we therefore develop a unified framework that is based on the orthogonal method but also cares extreme weights as in (Athey et al., 2018).

**Representation balancing and decomposition**  Starting with (Johansson et al., 2016; Shalit et al., 2017), a number of causal inference methods based on DNNs and representation balancing have been proposed (Li and Zhu, 2022). The representation-based methods have been demonstrated to be superior in complex problems such as nonlinear responses (Johansson et al., 2016), large actions spaces (Tanimoto et al., 2021) including continuous (Lopez et al., 2020) or structured spaces (Harada and Kashima, 2021), and so forth. These are end-to-end methods based on adversarial formulations. They virtually evaluate the worst-case with respect to the uncertainty of the model by distribution discrepancy between the representations of covariates in treated and control groups. On the other hand, representation balancing has certain limitations in an unrealistic theoretical assumption that the representation extractor should be invertible. It is shown that unobservable error terms arise when the invertibility is violated (Johansson et al., 2019; Zhao et al., 2019; Wu et al., 2020).

A solution to this problem is the representation decomposition (Hassanpour and Greiner, 2020; Wu et al., 2022; Cheng et al., 2022). They aim to identify confounding factors that affect both action selection and the outcomes and weights with only those factors. While it is an exciting approach, it is not guaranteed as a joint optimization. Joint optimization approaches have also been proposed for ATE estimation (Shi et al., 2019), though it may lead to cheating by less weighting to noisy regions especially under noise heterogeneity. Thus, we aim at a principled and versatile method without representation extraction while incorporating the advantages of end-to-end modeling by adversarial formulation.

**Distributionally robust optimization**  A similar framework aims at robust training concerning the deviation between the empirical and test distributions (Rahimian and Mehrotra, 2019). This approach and ours both account for distributional uncertainties and care for the worst possible cases in each setting. On the other hand, they do not suppose a significant difference between the training and test distributions but deal only with small perturbations. As a result, they do not face the main

problem that motivates this study, namely, extreme weighting by inverse probability. The uncertainty we seek to address appears in the density ratio of the test distribution to the training distribution, which can be roughly estimated from the association between background features and actual actions taken, but remains uncertain. Thus, although the problem and the approaches are similar, there is a significant difference from our setting.

**Pessimism in offline reinforcement learning**  Recent efforts in offline reinforcement learning revealed the benefit of pessimism on the candidate assessment (Rashidinejad et al., 2021; Buckman et al., 2021). In reinforcement learning, we consider modeling the cumulative expected reward in the long run as the Q function for assessing each action at each time step. The Q function is supposed to be maximized with respect to action $a$ during the inference phase. If the estimation error on an action is optimistic, i.e., if the Q value is overestimated, the action is likely to be selected over other better candidates. Therefore, conservative modeling of Q-function is preferable (Kumar et al., 2020), i.e., training a model to estimate below the true value when uncertain. The provable benefit of pessimism has been revealed in recent years (Buckman et al., 2021).

Our strategy also goes along with this direction. While offline reinforcement learning estimates the Q function and then optimizes it, we estimate the weighted loss and then optimize. We apply this pessimism principle to weighted estimation in causal inference; that is, our method pessimistically estimates the weighted loss with respect to the uncertainty of the weights.

## 4 NUISANCE-ROBUST WEIGHTING NETWORK

### 4.1 ADVERSARIAL REFORMULATION OF PLUG-IN NUISANCE

Inspired by pessimism in offline reinforcement learning, we build a causal inference method by the same principle. Most conventional causal inference methods employ a plug-in estimation approach: 1) estimate the nuisance propensity model $\hat{\mu}$ with its empirical evidence $\hat{E}$ (e.g., the cross entropy loss with the action as the label) and 2) plug it into the target empirical risk $\hat{L}$ (e.g., the MSE related to the CATE estimator $\hat{\tau}_\theta$)

$$\hat{\theta} = \arg\min_\theta \hat{L}(\theta; \hat{\mu}), \text{ where } \hat{\mu} = \arg\min_\mu \hat{E}(\mu) \tag{1}$$

as a substitute for the true propensity $\mu^* = \arg\min_\mu E(\mu)$. The specific form of the empirical loss $\hat{L}$ is discussed in Section 4.2.

The accuracy of the nuisance $\mu$ does not ultimately matter. Thus, it could be suboptimal to estimate and fix $\hat{\mu}$ without considering the subsequent estimation of the target parameter $\hat{\theta}$. we establish an error bound for any $\mu$, in which we utilize $\hat{E}$ as auxiliary evidence. Here, analogous to the effectiveness analysis of pessimism in offline reinforcement learning (Buckman et al., 2021), the sub-optimality compared to the true best parameter $\theta^* := \arg\min_\theta L(\theta; \mu^*)$ can be decomposed into optimistic/underestimation side and pessimistic/overestimation side.

**Theorem 4.1** (Adaptation from Theorem 1 in (Buckman et al., 2021)). *For any space $\Theta$, true population loss $L(\cdot; \mu^*)$, and proxy objective $\hat{J}(\cdot)$,*

$$L(\hat{\theta}; \mu^*) - L(\theta^*; \mu^*) \leq \inf_{\theta \in \Theta} \left\{ \hat{J}(\theta) - L(\theta^*; \mu^*) \right\} + \sup_{\theta \in \Theta} \left\{ L(\theta; \mu^*) - \hat{J}(\theta) \right\}, \tag{2}$$

*where $\hat{\theta} = \arg\min_{\theta \in \Theta} \hat{J}(\theta)$ and $\theta^* = \arg\min_{\theta \in \Theta} L(\theta; \mu^*)$.*

*Proof.* It follows from the definition of $\hat{\theta}$ and $\theta^*$. Details are given in Appendix A.  □

This analysis illustrates the asymmetry in the meaning of the estimation errors on the optimistic ($\hat{J}(\theta) < L(\theta; \mu^*)$) and pessimistic sides of the loss function ($\hat{J}(\theta) > L(\theta; \mu^*)$). The first term (2) corresponds to the estimation error on the pessimistic side and is infimum with respect to $\theta$, so it is sufficient if it is small for one $\theta$. On the other hand, the optimistic error in the second term should be uniformly small for all $\theta$. Unlike the pessimistic side, the optimistic error even for a single candidate can mislead the entire estimation process. This fact leads to the pessimism principle: "Be pessimistic

when uncertain." Following this principle, we focus on the optimistic error, i.e., the second term in (2), and enforce it to be bounded for all $\theta \in \Theta$.

Here, it would be instructive to see what optimism would result when we use the weighted risk as objective $\hat{J}(\theta) = \hat{L}(\theta; \mu)$ with a false propensity score $\mu$. The error is decomposed as follows.

$$L(\theta; \mu^*) - \hat{L}(\theta; \mu) = \underbrace{L(\theta; \mu^*) - L(\theta; \mu)}_{(a)} + \underbrace{L(\theta; \mu) - \hat{L}(\theta; \mu)}_{(b)} \tag{3}$$

The first term (a) is related to the uncertainty of the nuisance and can be majorized by maximizing the objective function with respect to $\mu$. Therefore, we define an uncertainty set $\mathcal{U}$ such that $\mu^*$ is included and take the worst case among them.

$$\mathcal{U} = \left\{ \mu \ \middle| \ \hat{E}(\mu) \leq \min_{\mu'} \hat{E}(\mu') + c \right\} \tag{4}$$

On the other hand, weights biased toward some samples for the pessimistic evaluation would decrease the effective sample size (Swaminathan and Joachims, 2015; Tanimoto et al., 2022). Thus we have to care about the statistical stability under $\mu$, which appears as (b) in (3). Swaminathan and Joachims (2015) proposed a pessimistic evaluation of logged bandits under true values of $\mu$, and Tanimoto et al. (2022) analyzed the sample complexity of linear class with weighted loss, both of which imply that we can majorize this term using the mean square of the weights. Though it depends on the class $\Theta$, we now assume there exists a majorizer $R(\mu) \geq$ (b), and let a stability constrained set $\mathcal{R}$ as

$$\mathcal{R} = \{ \mu \mid R(\mu) \leq C \} . \tag{5}$$

Now we can define our pessimistic loss as follows.

$$\hat{J}(\theta) = \max_{\mu \in \mathcal{U} \cap \mathcal{R}} \hat{L}(\theta; \mu), \tag{6}$$

where $\mathcal{U}$ is defined as (4) and $\mathcal{R}$ as (5). Our loss upper-bounds the optimistic evaluation error.

**Theorem 4.2** (Uniform bound with the pessimistic objective). *Suppose we have a model class $\Theta$ and a loss $L$ such that a stability constrained set $\mathcal{R}$ as (5) can be defined with $R$ so that $\hat{L}(\theta; \mu) - L(\theta; \mu) \leq R(\mu)$ for all $\theta \in \Theta$. Let the objective $\hat{J}$ defined as (6) with sufficiently large parameters $c$ and $C$ so that $\mu^* \in \mathcal{U} \cap \mathcal{R}$, i.e., $\hat{E}(\mu^*) \leq \min_{\mu'} \hat{E}(\mu') + c$ and $R(\mu^*) \leq C$. Then for all $\theta \in \Theta$ we have*

$$L(\theta; \mu^*) \leq \hat{J}(\theta) + C.$$

*Proof.* It follows from the definition of $\mathcal{U}, \mathcal{R}$ and $R$. Details can be found in Appendix Appendix A. □

Note that the nuisance uncertainty tolerance $c$ does not appear in this optimistic error upper bound. Being too pessimistic with large $c$ will lead to a large pessimistic-side error in (2). Therefore, $\mathcal{U}$ should be as small as possible while containing $\mu^*$.

## 4.2 Nuisance-Robust Transformed Outcome Regression Network

Under the aforementioned approach, next, we will discuss the application of our nuisance-robust loss to specific estimation methods. Among two-step methods of the form (1), a simple but clever method to directly estimate CATE is the transformed outcome regression (Athey and Imbens, 2016). This method estimates the target CATE without estimating each potential outcome. Based on pre-trained and fixed $\hat{\mu}$, the transformed outcome $z$ is defined as follows.

$$z^{(n)} = y_1^{(n)} \frac{a^{(n)}}{\hat{\mu}\left(x^{(n)}\right)} - y_0^{(n)} \frac{\left(1 - a^{(n)}\right)}{1 - \hat{\mu}\left(x^{(n)}\right)} \tag{7}$$

The transformed outcome is equivalent to CATE in expectation supposed that $\hat{\mu}$ is accurate: $\mathbb{E}[z|x] = \tau(x) = \mathbb{E}[Y_1 - Y_0|x]$ (see Appendix B for detail). Therefore, at the population level, the model $\hat{\tau}_\theta$

that minimizes MSE with respect to $z$ is consistent with that of minimizing PEHE. And, unlike CATE, $z$ can be calculated from observed values, since the unobserved potential outcome $y_{1-a}$ would be multiplied by $0$ and eliminated. We call this approach transformed outcome regression with propensity weighting (PW), or PWNet in short.

While we do not suffer from estimation errors in potential outcomes, $z$ in PWNet (7) has a large variance in general. Therefore, we can also utilize pre-trained potential outcome models $\hat{f}_a(x) \simeq \mathbb{E}[Y_a|x]$ to reduce the variance in the transformed outcome. Kennedy (2020) proposed using the plug-in CATE estimate $\hat{f}_1(x^{(n)}) - \hat{f}_0(x^{(n)})$ with residual adjusted with weights. Despite being a plug-in method, the estimation error of these models $(f_1, f_0)$ does not necessarily ruin the later step. If either $(f_1, f_0)$ or $\mu$ is accurate, $z$ works as an accurate target, which is called the double robustness (Kennedy, 2020). This method is called the doubly robust learner or DRNet.

Taking DRNet as a baseline plug-in method, we formulate our proposed method nuisance-robust transformed outcome regression, NuNet. Our transformed outcome $z_\mu$ with mutable nuisance $\mu$ is defined as follows.

$$
z_\mu^{(n)} = \hat{f}_1(x^{(n)}) - \hat{f}_0(x^{(n)}) + \frac{y_1^{(n)} - \hat{f}_1(x^{(n)})}{\mu(x^{(n)})} a^{(n)} - \frac{y_0^{(n)} - \hat{f}_0(x^{(n)})}{1 - \mu(x^{(n)})}(1 - a^{(n)}). \tag{8}
$$

We start with a pre-trained nuisance $\mu_0$ and randomly initialized parameter $\theta$ and perform adversarial optimization of them as in (6) with the MSE $\hat{L}(\theta; \mu) = \frac{1}{N} \sum_n \left( z_\mu^{(n)} - \tau_\theta(x^{(n)}) \right)^2$. While DRNet uses pre-trained and fixed nuisance models $\hat{f}_1, \hat{f}_0$, and $\hat{\mu}$ for training $\hat{\tau}_\theta$, we utilize pre-trained $\mu_0$ for initialization but update simultaneously with $\theta$. Although $f_0$ and $f_1$ are also nuisance parameters, the uncertainty of them do not differ among the target parameter space, thus we need not take care. Excluding the error correction term in (8), a disturbance $(\Delta f_0, \Delta f_1)$ can be absorbed by translation $\tau \leftarrow \tau + \Delta f_1 - \Delta f_0$.

## 4.3 Implementation of Constraints for Gradient-based Update

We defined the evidence level $\hat{E}(\mu)$ and the statistical stability $R(\mu)$ as constraints in Section 4.1. Typical constraint implementation is as regularization term such as $\alpha \max\{0, \hat{E}(\mu) - \hat{E}(\mu_0) - c\}$, where $\mu_0$ is the pre-trained and fixed propensity model. However, since $-\hat{L}(\theta; \mu)$ is not convex with respect to $\mu$, the regularization cannot reproduce the constrained optimization within $\mathcal{U}$. $\mu$ would diverge to extreme values to increase $\hat{L}(\theta; \mu)$ with no regard for the increase in the regularization term. Therefore, for $\hat{E}$, we employ a (simplified version of) augmented Lagrangian method (Bertsekas, 2014). We use the regularization term $\alpha_k \max\{0, \hat{E}(\mu) - \hat{E}(\mu_0) - c\}$, where $\alpha_k = \alpha_0 \gamma^k$ with the training epoch $k$. Note that the hyperparameters $\alpha_0$ and $\gamma$ are for optimization and could be tuned only by the training data. The only statistical hyperparameter here is the tolerance $c$, which is to be selected by validation.

For the statistical stability $R$, we employ squared weights simply as a regularization term. Let $w_\mu^{(n)} = \frac{a^{(n)}}{\mu(x^{(n)})} + \frac{1 - a^{(n)}}{1 - \mu(x^{(n)})}$ the weight under the nuisance function $\mu$. Finally, our adversarial objective at the $k$-th iteration is the following.

$$
\hat{J}(\theta, \mu) = \frac{1}{N} \sum_n \left( z_\mu^{(n)} - \tau_\theta(x^{(n)}) \right)^2 - \alpha_k \max\{0, \hat{E}(\mu) - \hat{E}(\mu_0) - c\} - \beta \frac{1}{N} \sum_n \left( w_\mu^{(n)} \right)^2. \tag{9}
$$

In each iteration, we minimize $\hat{J}(\theta, \mu)$ with respect to $\theta$ and maximize with respect to $\mu$. Overall, our loss controls the optimistic side error due to the uncertainty of $\mu$ (3-a) by maximizing the first term with respect to $\mu$ under the likelihood constraint in the second term, while simultaneously controlling the optimistic side error of the weighted empirical loss (3-b) by flattening the weight with the third term. The whole algorithm is summarized in Algorithm 1 and Figure 1 illustrates the architecture.

---

**Algorithm 1** Nuisance-Robust Transformed Outcome Regression Network (NuNet)

---

**Input:** Training data $D = \{(x^{(n)}, a^{(n)}, y^{(n)})\}_n$, hyperparameters $\alpha_0, \gamma, \beta, c$, validation ratio $r$
**Output:** Trained network parameter $\theta$ and validation error
 1: Train $f_1, f_0, \mu$ by an arbitrary supervised learning method, e.g.:
    $\hat{f}_a \leftarrow \arg\min_{f_a} \frac{1}{N} \sum_{n:a^{(n)}=a} (y^{(n)} - f_a(x^{(n)}))^2$ for each $a \in \{0, 1\}$,
    $\mu_0 \leftarrow \arg\min_\mu -\frac{1}{N} \sum_n a^{(n)} \log \mu(x^{(n)}) + (1 - a^{(n)}) \log(1 - \mu(x^{(n)}))$
 2: Split train and validation $D_{\mathrm{tr}}, D_{\mathrm{val}}$ by the ratio $r$
 3: $k \leftarrow 0$
 4: **while** Convergence criteria is not met **do**
 5:     **for** each sub-sampled mini-batch from $D_{\mathrm{tr}}$ **do**
 6:         Update parameters with objective (9) and step sizes $\eta_\theta, \eta_\mu$ from optimizes:
 7:         $\theta_{k+1} \leftarrow \theta_k - \eta_\theta \nabla_\theta \hat{J}(\theta, \mu)$
 8:         $\mu_{k+1} \leftarrow \mu_k + \eta_\mu \nabla_\mu \hat{J}(\theta, \mu)$
 9:         $\alpha_{k+1} \leftarrow \gamma \alpha_k$
10:         $k \quad \leftarrow k + 1$
11:     Check convergence criterion with validation error $\frac{1}{N_{\mathrm{val}}} \sum_{n \in D_{\mathrm{val}}} (z_{\mu_0}^{(n)} - \tau_\theta(x^{(n)}))^2$
12: **return** $\theta$ and the last validation error for model selection

---

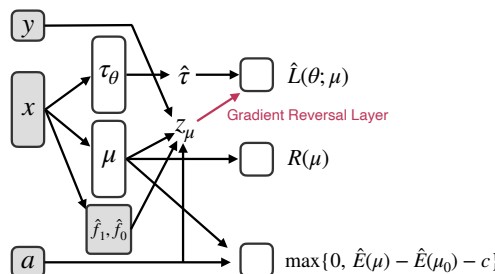

Figure 1: The architecture of NuNet. Gray boxes are fixed during the training. The nuisance function $\mu$ is trained to maximize the empirical loss $\hat{L}(\theta; \mu)$ while minimizing the other terms. This adversarial formulation can be expressed as joint minimization with the gradient reversal layer.

## 5 EXPERIMENT

### 5.1 SETTINGS

**Synthetic data generation** We followed the setup of (Curth and van der Schaar, 2021b) for synthetic data generation processes and model hyperparameters, except for our noise heterogeneous setting. We generated $d = 25$ dimensional multivariate normal covariates including 5 confounding factors that affect outcome and treatment selection, 5 outcome-related factors, and 5 purely CATE-related factors. The true outcome and CATE models were nonlinear. Other details are described in Appendix C.

In addition to the original additive noise (AN) setting $y = y_{\mathrm{true}} + \varepsilon$, we also tested on the multiplicative noise (MN) setting $y = y_{\mathrm{true}}(1 + \varepsilon)$, where $\varepsilon$ is drawn from a normal distribution with its average noise level was adjusted to the one in AN. Most causal inference literature uses synthetic data or semi-synthetic data where only covariates are real and synthesizes outcomes with additive noise, which do not reflect the heterogeneity of the real environment. Noise heterogeneity is critical since the optimistic-side error will likely emerge, and naive joint optimization suffers. We therefore set this up as a simple data generation model with heterogeneous noise.

**Real-world dataset** Most well-established semi-synthetic datasets have real-world covariates and synthesized outcomes and do not reproduce up to noise heterogeneity, while the Twins dataset from (Almond et al., 2005) had both potential outcomes recorded. The first-year mortality of twins born at low birth weights was treated as potential outcomes for the heavier and lighter-born twins, respectively, and weighted subsampling was performed to reproduce the missing counterfactuals. The test

Table 1: PEHE on additive noise dataset (mean $\pm$ standard error on 10 runs). The best results are shown in bold, and comparable results are italicized and underlined.

| Method | N=500 | 1000 | 2000 | 5000 | 10000 | 20000 |
|--------|-------|------|------|------|-------|-------|
| TNet | $18.55 \pm 0.88$ | $13.89 \pm 1.10$ | $5.02 \pm 0.14$ | $1.96 \pm 0.06$ | $1.22 \pm 0.03$ | $0.88 \pm 0.02$ |
| TARNet | $18.21 \pm 1.12$ | $8.77 \pm 0.35$ | $4.28 \pm 0.19$ | $1.74 \pm 0.06$ | $1.06 \pm 0.02$ | $0.76 \pm 0.03$ |
| CFR | $17.90 \pm 1.18$ | $8.77 \pm 0.35$ | $4.28 \pm 0.19$ | $1.71 \pm 0.05$ | $1.05 \pm 0.02$ | $0.76 \pm 0.03$ |
| SNet3 | $\mathbf{13.10 \pm 0.65}$ | $\underline{\mathit{7.73 \pm 0.34}}$ | $3.85 \pm 0.11$ | $1.54 \pm 0.05$ | $0.99 \pm 0.02$ | $0.62 \pm 0.01$ |
| SNet | $\underline{\mathit{14.14 \pm 0.57}}$ | $\mathbf{7.17 \pm 0.29}$ | $\mathbf{3.39 \pm 0.11}$ | $\mathbf{1.26 \pm 0.03}$ | $\mathbf{0.74 \pm 0.02}$ | $\mathbf{0.43 \pm 0.01}$ |
| PWNet | $18.46 \pm 0.82$ | $13.03 \pm 0.54$ | $15.97 \pm 0.68$ | $20.99 \pm 1.25$ | $25.31 \pm 2.32$ | $19.21 \pm 1.36$ |
| DRNet | $16.56 \pm 0.75$ | $11.58 \pm 0.66$ | $3.91 \pm 0.14$ | $1.45 \pm 0.04$ | $1.14 \pm 0.11$ | $0.66 \pm 0.03$ |
| NuNet | $15.78 \pm 0.69$ | $11.43 \pm 0.48$ | $4.02 \pm 0.09$ | $1.52 \pm 0.07$ | $0.86 \pm 0.01$ | $0.54 \pm 0.01$ |

target is the difference between the noisy potential outcomes $\tau^{(n)} = y_1^{(n)} - y_0^{(n)}$ instead of CATE $\mathbb{E}[\tau^{(n)}|x]$. We followed (Curth and van der Schaar, 2021a) for sampling strategy and other details.

Also, the Jobs dataset from (LaLonde, 1986) has a randomized test set based on an experimental job training program and an observational training set. Although we do not have both potential outcomes, we can substitute the CATE in PEHE with the transformed outcome since the true propensity is known as constant in the randomized test set, as proposed in (Curth et al., 2021). A detailed justification for this evaluation can be found in Appendix B. Most popular semi-synthetic datasets used in causal inference evaluation have generated outcomes and do not reproduce noise heterogeneity in real situations. That is why we used these datasets with real outcomes.

**Baseline methods** We compared several representative methods for causal inference with DNN. TNet was a simple plug-in method that estimates each potential outcome with separate two networks and then outputs the difference between their predictions. SNet and SNet3 were decomposed representation-based methods that shared representation extractors for outcome and propensity estimation layers. They have three kinds of extractors, namely, outcome-only, propensity-only, and shared representation for confounders. SNet3 was a modified version, not using weighting in (Curth and van der Schaar, 2021b), of what was originally proposed as DR-CFR (Hassanpour and Greiner, 2019) and DeR-CFR (Wu et al., 2022) for simultaneous optimization of the weights and the outcome model. SNet was an even more flexible architecture than SNet3 that had shared and independent extractors for each potential outcome, proposed in (Curth and van der Schaar, 2021b). PWNet and DRNet were DNN implementations of transformed outcome methods (Athey and Imbens, 2016; Kennedy, 2020), in which networks in the first step were independent for each potential outcome and propensity. In the second step, a single CATE network was trained. Our proposed NuNet is based on DRNet architecture.

**Hyparparameters and model selection** We set the candidates of the hyperparameters as $\alpha_0 \in \{1, 10\}$, $\gamma \in \{1.01, 1.03, 1.05\}$, and $\beta \in \{10, 100, 300\}$. For the experiment with Twins, we fixed them as $\alpha_0 = 10, \gamma = 1.03, \beta = 100$. The tolerance was fixed at $c = 0.03$. Model selection, including early stopping, was performed on the evidence measure of each method with 30% of the training data held out for validation. That is, for transformed outcome methods, we evaluated the MSE with respect to the transformed outcome $z$ in each method. For our proposed NuNet, we used the pre-trained and fixed transformed outcome for validation, just as in DRNet.

## 5.2 RESULTS

**AN setting** The results are shown in Table 1. Our proposed NuNet outperformed or was at least comparable to, DRNet, the baseline plug-in method of NuNet. On the other hand, representation-based methods (SNet3 and SNet) outperformed the transformed outcome methods (PWNet, DRNet, and NuNet). The shared representation extractor of the confounding factors could be an effective inductive bias, especially with small sample sizes. SNet is overall more accurate than SNet3 since it can also share parameters for components common to both potential outcomes.

**MN setting** Table 2 shows the results in PEHE. Our proposed method outperformed other baselines when the sample size was relatively sufficient. Unlike the AN setting, the multiplicative (heterogeneous) noise setting tends to be optimistic. The pessimistic evaluation with more emphasis on

Table 2: PEHE on multiplicative noise dataset

| Method | N=500 | 1000 | 2000 | 5000 | 10000 | 20000 |
|---|---|---|---|---|---|---|
| TNet | $22.03 \pm 1.23$ | $17.59 \pm 0.89$ | $11.97 \pm 0.40$ | $5.93 \pm 0.16$ | $3.76 \pm 0.08$ | $2.52 \pm 0.11$ |
| TARNet | $20.75 \pm 0.99$ | $\mathbf{13.06 \pm 0.53}$ | *$10.25 \pm 0.36$* | $5.27 \pm 0.18$ | $3.17 \pm 0.08$ | $2.10 \pm 0.09$ |
| CFR | $20.30 \pm 1.05$ | $\mathbf{13.06 \pm 0.53}$ | $\underline{10.10 \pm 0.32}$ | $5.22 \pm 0.18$ | $3.16 \pm 0.08$ | $2.07 \pm 0.09$ |
| SNet3 | $\mathbf{17.83 \pm 0.94}$ | $15.44 \pm 0.65$ | $11.12 \pm 0.36$ | $5.71 \pm 0.25$ | $3.61 \pm 0.14$ | $2.46 \pm 0.09$ |
| SNet | *$18.44 \pm 0.86$* | $15.73 \pm 0.58$ | $11.22 \pm 0.33$ | $5.47 \pm 0.17$ | $3.12 \pm 0.08$ | $2.01 \pm 0.07$ |
| PWNet | *$18.97 \pm 0.90$* | *$13.14 \pm 0.54$* | $15.95 \pm 0.64$ | $21.08 \pm 1.22$ | $25.63 \pm 2.31$ | $20.92 \pm 2.09$ |
| DRNet | $19.96 \pm 1.01$ | $15.34 \pm 0.75$ | *$9.93 \pm 0.40$* | *$4.80 \pm 0.21$* | $3.20 \pm 0.24$ | $1.83 \pm 0.10$ |
| NuNet | *$19.96 \pm 1.20$* | $15.54 \pm 0.57$ | $\mathbf{9.83 \pm 0.45}$ | $\mathbf{4.67 \pm 0.32}$ | $\mathbf{2.44 \pm 0.09}$ | $\mathbf{1.50 \pm 0.06}$ |

Table 3: MSE for noisy CATE on real-world datasets (mean $\pm$ standard error on 5 runs).

| Method | Twins | | | | | Jobs |
| | N=500 | 1000 | 2000 | 5000 | 11400 | N=2570 |
|---|---|---|---|---|---|---|
| TNet | $0.354 \pm .004$ | $0.350 \pm .002$ | $0.329 \pm .001$ | $0.324 \pm .002$ | $0.322 \pm .001$ | $9.42 \pm .12$ |
| TARNet | $0.336 \pm .003$ | $0.336 \pm .002$ | $0.326 \pm .001$ | *$0.320 \pm .001$* | *$0.321 \pm .001$* | $9.33 \pm .02$ |
| CFR | $\mathbf{0.322 \pm .001}$ | *$0.324 \pm .002$* | *$0.323 \pm .002$* | *$0.321 \pm .001$* | *$0.321 \pm .001$* | $9.33 \pm .02$ |
| SNet3 | $0.331 \pm .002$ | $0.330 \pm .002$ | *$0.322 \pm .001$* | $\mathbf{0.319 \pm .001}$ | *$0.320 \pm .001$* | $9.38 \pm .06$ |
| SNet | $0.333 \pm .002$ | $0.331 \pm .002$ | *$0.323 \pm .001$* | *$0.320 \pm .001$* | *$0.320 \pm .001$* | $9.36 \pm .06$ |
| PWNet | $0.330 \pm .001$ | $0.327 \pm .001$ | $0.324 \pm .001$ | *$0.322 \pm .002$* | $0.323 \pm .001$ | $8.82 \pm .03$ |
| DRNet | $0.340 \pm .002$ | $0.338 \pm .001$ | *$0.322 \pm .001$* | $0.323 \pm .001$ | $0.323 \pm .001$ | $9.10 \pm .02$ |
| NuNet | $0.326 \pm .002$ | $\mathbf{0.323 \pm .002}$ | $\mathbf{0.320 \pm .001}$ | *$0.321 \pm .001$* | $\mathbf{0.319 \pm .001}$ | $\mathbf{8.62 \pm .06}$ |

hard instances would be a reasonable explanation for the superiority of the proposed method. Even though representation decomposition should also be useful in the MN setting since the data generation model was the same as the AN setting except for noise, the weighting approach was superior to the representation decomposition method without weighting. Combining representation decomposition and weighting is a promising future direction. Simultaneous optimization is required and our approach based on plug-in (multi-step) baseline methods cannot be simply applied.

**Real-world datasets**   Experiments on Twins data also showed the superiority of NuNet in most cases as in Table 3. Note that the test target $\tau^{(n)}$ is noisy, and the value contains the noise variance. In the small sample cases as $N = 500$, the methods with fixed outcome models (TNet and DR-Net) underperformed compared to the methods without outcome models or shared representations. NuNet based on DRNet should suffer the same disadvantage, though it still showed superior performance to those baseline methods. Table 3 also showed the results on Jobs data, which exhibits similar trends. Although the accuracy of PWNet is relatively high, probably due to the low dimensionality of $d = 8$, the proposed DRNet-based method still outperformed other baseline methods. Note that the evaluation metric for Jobs, the MSE with respect to the transformed outcome, also contains constant noise.

## 6   CONCLUSION

We proposed NuNet to unify the two steps of nuisance estimation and target estimation in a single step based on the pessimism principle. We have empirically demonstrated that existing methods not based on weighting and methods based on weighting with a two-step strategy tend to be optimistic, and the proposed method exhibited superior performances, especially in noise heterogeneity. To the best of our knowledge, our approach is the first attempt at a principled solution based on pessimism, and it sheds light on the approach for making various multi-step inference methods end-to-end.

**Limitation and future work**   Our framework has wide potential applications to plug-in methods, not only the proposed method. On the other hand, it cannot be applied to methods that are originally formulated as joint optimization of models and weights, such as representation decomposition. Such models that have representation shared by the model and weights often exhibit effective inductive bias, which is another aspect of the combination of the weighting approach and DNNs. Deriving a pessimism-based theoretical framework for such methods and investigating principled learning methods would be a promising future direction for versatile causal inference methods.

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
