.001$ | $\mathit{0.320 \pm .001}$ | $\mathit{0.321 \pm .001}$ | $9.33 \pm .02$ |
| CFR | $\mathbf{0.322 \pm .001}$ | $\mathit{0.324 \pm .002}$ | $\mathit{0.323 \pm .002}$ | $\mathit{0.321 \pm .001}$ | $\mathit{0.321 \pm .001}$ | $9.33 \pm .02$ |
| SNet3 | $0.331 \pm .002$ | $0.330 \pm .002$ | $\mathit{0.322 \pm .001}$ | $\mathbf{0.319 \pm .001}$ | $\mathit{0.320 \pm .001}$ | $9.38 \pm .06$ |
| SNet | $0.333 \pm .002$ | $0.331 \pm .002$ | $\mathit{0.323 \pm .001}$ | $\mathit{0.320 \pm .001}$ | $\mathit{0.320 \pm .001}$ | $9.36 \pm .06$ |
| PWNet | $0.330 \pm .001$ | $0.327 \pm .001$ | $0.324 \pm .001$ | $\mathit{0.322 \pm .002}$ | $0.323 \pm .001$ | $8.82 \pm .03$ |
| DRNet | $0.340 \pm .002$ | $0.338 \pm .001$ | $\mathit{0.322 \pm .001}$ | $0.323 \pm .001$ | $0.323 \pm .001$ | $9.10 \pm .02$ |
| NuNet | $0.326 \pm .002$ | $\mathbf{0.323 \pm .002}$ | $\mathbf{0.320 \pm .001}$ | $\mathit{0.321 \pm .001}$ | $\mathbf{0.319 \pm .001}$ | $\mathbf{8.62 \pm .06}$ |

hard instances would be a reasonable explanation for the superiority of the proposed method. Even though representation decomposition should also be useful in the MN setting since the data generation model was the same as the AN setting except for noise, the weighting approach was superior to the representation decomposition method without weighting. Combining representation decomposition and weighting is a promising future direction. Simultaneous optimization is required and our approach based on plug-in (multi-step) baseline methods cannot be simply applied.

**Real-world datasets** Experiments on Twins data also showed the superiority of NuNet in most cases as in Table 3. Note that the test target $\tau^{(n)}$ is noisy, and the value contains the noise variance. In the small sample cases as $N = 500$, the methods with fixed outcome models (TNet and DRNet) underperformed compared to the methods without outcome models or shared representations. NuNet based on DRNet should suffer the same disadvantage, though it still showed superior performance to those baseline methods. Table 3 also showed the results on Jobs data, which exhibits similar trends. Although the accuracy of PWNet is relatively high, probably due to the low dimensionality of $d = 8$, the proposed DRNet-based method still outperformed other baseline methods. Note that the evaluation metric for Jobs, the MSE with respect to the transformed outcome, also contains constant noise.

## 6 CONCLUSION

We proposed NuNet to unify the two steps of nuisance estimation and target estimation in a single step based on the pessimism principle. We have empirically demonstrated that existing methods not based on weighting and methods based on weighting with a two-step strategy tend to be optimistic, and the proposed method exhibited superior performances, especially in noise heterogeneity. To the best of our knowledge, our approach is the first attempt at a principled solution based on pessimism, and it sheds light on the approach for making various multi-step inference methods end-to-end.

**Limitation and future work** Our framework has wide potential applications to plug-in methods, not only the proposed method. On the other hand, it cannot be applied to methods that are originally formulated as joint optimization of models and weights, such as representation decomposition. Such models that have representation shared by the model and weights often exhibit effective inductive bias, which is another aspect of the combination of the weighting approach and DNNs. Deriving a pessimism-based theoretical framework for such methods and investigating principled learning methods would be a promising future direction for versatile causal inference methods.

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

## A  PROOFS

**Theorem A.1** (Adaptation from Theorem 1 in ). *For any space $\Theta$, true population loss $L(\cdot; \mu^*)$, and proxy objective $\hat{J}(\cdot)$,*

$$L(\hat{\theta}; \mu^*) - L(\theta^*; \mu^*) \leq \inf_{\theta \in \Theta} \left\{ \hat{J}(\theta) - L(\theta^*; \mu^*) \right\} + \sup_{\theta \in \Theta} \left\{ L(\theta; \mu^*) - \hat{J}(\theta) \right\},$$

*where $\hat{\theta} = \arg\min_{\theta \in \Theta} \hat{J}(\theta)$ and $\theta^* = \arg\min_{\theta \in \Theta} L(\theta; \mu^*)$.*

*Proof.*

$$L(\hat{\theta}; \mu^*) - L(\theta^*; \mu^*) = \inf_{\theta \in \Theta} \left\{ L(\hat{\theta}; \mu^*) - \hat{J}(\hat{\theta}) + \underbrace{\hat{J}(\hat{\theta}) - \hat{J}(\theta)}_{\leq 0 \ \forall \theta \in \Theta} + \hat{J}(\theta) - L(\theta^*; \mu^*) \right\}$$

$$\leq L(\hat{\theta}; \mu^*) - \hat{J}(\hat{\theta}) + \inf_{\theta \in \Theta} \left\{ \hat{J}(\theta) - L(\theta^*; \mu^*) \right\}$$

$$\leq \sup_{\theta \in \Theta} \left\{ L(\theta; \mu^*) - \hat{J}(\theta) \right\} + \inf_{\theta \in \Theta} \left\{ \hat{J}(\theta) - L(\theta^*; \mu^*) \right\}$$

$\square$

**Theorem A.2** (Uniform bound with the pessimistic objective). *Suppose we have a model class $\Theta$ and a loss $L$ such that a stability constrained set $\mathcal{R}$ as (5) can be defined with $R$ so that $\hat{L}(\theta; \mu) - L(\theta; \mu) \leq R(\mu)$ for all $\theta \in \Theta$. Let the objective $\hat{J}$ defined as (6) with sufficiently large parameters $c$ and $C$ so that $\mu^* \in \mathcal{U} \cap \mathcal{R}$, i.e., $\hat{E}(\mu^*) \leq \min_{\mu'} \hat{E}(\mu') + c$ and $R(\mu^*) \leq C$. Then for all $\theta \in \Theta$ we have*

$$L(\theta; \mu^*) \leq \hat{J}(\theta) + C.$$

*Proof.*

$$L(\theta; \mu^*) - \hat{J}(\theta) = L(\theta; \mu^*) - \max_{\mu \in \mathcal{U} \cap \mathcal{R}} \left\{ \hat{L}(\theta; \mu) - L(\theta; \mu) + L(\theta; \mu) \right\}$$

$$\leq L(\theta; \mu^*) - \max_{\mu \in \mathcal{U} \cap \mathcal{R}} \left\{ -C + L(\theta; \mu) \right\} \qquad \text{(Def. of } R \text{ and } \mathcal{R}\text{)}$$

$$\leq L(\theta; \mu^*) + C - L(\theta; \mu^*) = C \qquad (\mu^* \in \mathcal{U} \cap \mathcal{R})$$

$\square$

## B  EQUIVALENCE BETWEEN THE TRANSFORMED OUTCOME AND THE CATE IN EXPECTATION

Our empirical risk and the evaluation criteria are based on the method of transformed outcome . The IPW transformed outcome $z$ in (7) is equivalent to the CATE in the sense of its conditional expectation:

$$\mathbb{E}[z|x] = \mathbb{E}_{Y_0, Y_1, A \sim \mu(x)} \left[ Y_1 \frac{A}{\mu(x)} - Y_0 \frac{1 - A}{1 - \mu(x)} \middle| x \right]$$

$$= \mathbb{E}[Y_1 - Y_0|x] =: \tau(x).$$

Then, letting $z = \tau(x) + \varepsilon$ with $\mathbb{E}[\varepsilon|x] = 0$, we have

$$\mathbb{E}_{z,x}[(z - \hat{\tau})^2] = \mathbb{E}_{\varepsilon,x}[(\tau(x) + \varepsilon - \hat{\tau})^2]$$

$$= \mathbb{E}_x[(\tau(x) - \hat{\tau})^2] + 2\mathbb{E}_x[\mathbb{E}_\varepsilon[\varepsilon|x](\tau(x) - \hat{\tau})] + \mathbb{E}_{\varepsilon,x}[\varepsilon^2]$$

$$= \text{PEHE} + \mathbb{E}_x \mathbb{V}[\varepsilon].$$

The MSE on $z$ is equivalent to our final metric PEHE except for a constant term. The same equivalence can be derived for the doubly robust transformed outcome in (8). This equivalence justifies our employed MSE on $z$ as the empirical risk $\hat{L}$ and the evaluation metric.

# C EXPERIMENTAL DETAILS

## C.1 SIMULATION ENVIRONMENT

Our synthetic data in the additive noise (AN) setting was identical to the setting used in , which was inspired by the decomposed covariate setting used in . We used $d = 25$ dimensional normal covariates $x$. Out of 25 covariates, there were $d_o = 5$ outcome-related covariates $x_o$ that affect only potential outcomes, $d_c = 5$ confounders $x_c$ that affect both potential outcomes and treatment assignment, and $d_t = 5$ covariates that affect treatment effect $x_t$.

The expected potential outcomes were calculated as follows.

$$\mathbb{E}[Y_0|x] = \mathbf{1}^\top \begin{bmatrix} x_c \\ x_o \end{bmatrix}^2,$$

$$\mathbb{E}[Y_1|x] = \mathbb{E}[Y_0|x] + \tau(x),$$

Squaring works on an element-by-element basis where $\mathbf{1} = [1, \cdots, 1]^\top$, squaring $\cdot^2$ is element-wise, and the treatment effect $\tau(x)$ was defined as

$$\tau(x) = \mathbf{1}^\top x_t^2.$$

The true propensity that affects the treatment assignment was defined as

$$\mu(x) = \mu(a = 1|x) = \sigma(\xi(\mathbf{1}^\top x_c^2/d_c - \omega)),$$

where $\sigma$ was the sigmoid, $\xi$ was the strength of selection and was set as $\xi = 3$, and $\omega$ was adaptively set so that the median of the inside $\sigma$ would be 0.

The expected factual outcome is defined as follows.

$$\bar{y} = A\mathbb{E}[Y_1|x] + (1 - A)\mathbb{E}[Y_0|x]$$

where $A \sim \text{Bernoulli}(\mu(A = 1|x))$. In the AN setting, the outcome was observed with additive noise as

$$y = \bar{y} + \varepsilon,$$

where $\varepsilon \sim \mathcal{N}(0, 1)$. In the multiplicative noise (MN) setting, it was

$$y = \bar{y}(1 + \varepsilon'),$$

where $\varepsilon' \sim \mathcal{N}(0, \xi)$ with its standard deviation $\xi = 2 / \left( \sqrt{\text{Var}[\mathbb{E}[Y_1|x]]} + \sqrt{\text{Var}[\mathbb{E}[Y_0|x]]} \right)$.

## C.2 ARCHITECTURE AND HYPERPARAMETERS

**Synthetic data experiment** We followed the implementation of for the synthetic data experiment. We employed the multi-layer perceptron with representation (input-side) layers and hypothesis (output-side) layers. For the separated models (TNet, PWNet, DRNet, and NuNet), the representation layers were 3 layers with 200 units each and the hypothesis layers were 2 layers with 100 units each, for each prediction of $y_0$, $y_1$, and $a$. SNet3 had 3 representations of outcome-only (50 units $\times$ 3 layers), treatment-only (50 units $\times$ 3 layers), and shared representation layers (150 units $\times$ 3 layers). SNet had 5 representations for outcome related 3 representations ($y_1$-only, $y_0$-only, and outcome-shared, of 50 units $\times$ 3 layers), treatment-only (100 units $\times$ 3 layers), and shared for outcomes and treatment (100 units $\times$ 3 layers). We used the exponential linear unit (ELU) activations  and the optimizer was Adam . For NuNet, we applied model selection for the training epochs with the same validation data due to the instability of adversarial training. NuNet sometimes diverges within the minimum training epochs or the early-stopping patience epochs of 10. Therefore, it would be better to keep the best-so-far parameters in each epoch and output it.

**Real-world dataset experiment** We followed  for the Twins experiment. The representation layers and the hypothesis layers were only 1 in all methods. For NuNet, instead of model selection of epochs, shorter minimum epochs of 40 were used to avoid overfitting the pre-trained $\mu$, as opposed to 200 in other methods as in .

**Infrastructure** All the experiments were run on a machine with 28 CPUs (Intel(R) Xeon(R) CPU E5- 2680 v4 @ 2.40GHz), 250GB memory, and 8 GPUs (NVIDIA GeForce GTX 1080 Ti).