# OpenReview forum: "Nuisance-Robust Weighting Network for End-to-End Causal Effect Estimation"
_ICLR.cc/2024/Conference — Submitted to ICLR 2024_

### Official Review · Reviewer_KLj8 · 2023-11-04

**Soundness:** 3 good
**Presentation:** 3 good
**Contribution:** 4 excellent
**Rating:** 8
**Confidence:** 2

**Summary:**

The research introduces the Nuisance-Robust Transformed Outcome Regression Network (NuNet) within a standard causal inference framework, which aims to discern between factual and counterfactual potential outcomes using observational data. In empirical tests, many established methods showcased an inclination towards optimism particularly under conditions of noise heterogeneity. NuNet distinguishes itself by merging nuisance estimation and target estimation into a singular step, guided by the pessimism principle. The primary goal is to pinpoint a potential outcome function to determine the causal eﬀect of a treatment action. Accuracy is gauged through the PEHE, contingent upon three foundational assumptions: the Stable Unit Treatment Value, unconfoundedness, and overlap.

Empirical tests claim that NuNet oLen surpasses or parallels baseline plug-in methods, particularly in diverse noise seMngs and real-world datasets. However, it faces challenges with techniques prioritizing joint optimization. Inspired by pessimism in oﬄine reinforcement learning, this causal inference method oﬀers a diﬀerent approach. Most conventional techniques adopt a plug-in estimation, but this may show sub-optimality if nuisance accuracy isn't accounted for. The signiﬁcance of addressing the gap between optimistic and pessimistic errors is emphasized, leading to adherence to the pessimism principle. The study also delves into various methods used for estimating the CATE. Notable methods include Transformed Outcome Regression, PWNet, Doubly Robust Learner (DRNet), and NuNet. These strategies aim to enhance robustness against nuisances and unobserved confounders, ultimately aiming for reliable and accurate estimations. For future exploration, the authors suggested constructing a pessimism-based theoretical structure and delving into principled learning avenues could propel the evolution of causal inference methodologies.

**Strengths:**

1. Clear mention of shortcomings of the conventional statistical and end-to-end adversarial learning networks.
2. Evaluation of robustness of the approach under conditions including heterogeneous noise, AN setting, MN setting and real-world datasets.

**Weaknesses:**

Considerable diﬀerence between the results of NuNet and SNet (best performing) pertaining to PEHE on the additive noise dataset.

**Questions:**

See Weaknesses.

---

> ### Author Response · Authors · 2023-11-15
>
> Dear reviewer KLj8, many thanks for your positive feedback and for concerning the empirical comparison.
>
> ## Q1. Considerable diﬀerence between the results of NuNet and SNet (best performing) pertaining to PEHE on the additive noise dataset.
>
> Compared to SNet, our base model DRNet is also outpaced in the AN setting (NuNet generally outperforms DRNet).
> SNet has a special network architecture designed for CATE estimation (shared representation for $f$s and $\mu$), which may serve as an effective inductive bias.
> The exploration of architecture is an important aspect that is orthogonal to our considerations.
> Combining such an architecture is an important topic for future work as we wrote in the conclusion.
>
> > On the other hand, it cannot be applied to methods that are originally
> formulated as joint optimization of models and weights, such as representation decomposition. Such
> models that have representation shared by the model and weights often exhibit effective inductive
> bias, which is another aspect of the combination of the weighting approach and DNNs. Deriving
> a pessimism-based theoretical framework for such methods and investigating principled learning
> methods would be a promising future direction for versatile causal inference methods.
>
> Again, thank you for your time and enthusiasm in reading our paper and providing feedback.

---

### Official Review · Reviewer_S5uy · 2023-11-09

**Soundness:** 2 fair
**Presentation:** 3 good
**Contribution:** 2 fair
**Rating:** 5
**Confidence:** 4

**Summary:**

The paper proposes a new method for estimating CATE. In contrast to previous two-stage estimators, the propensity score is simultaneously optimized with the second-stage regression. This is done in an adversarial manner to ensure robustness regarding estimation errors in the propensity score. The method is evaluated using simulated and real-world data.

**Strengths:**

- The paper is well written.
- CATE estimation is an important problem, with applications in various domains.
- The proposed method performs well empirically.

**Weaknesses:**

- I am not convinced of the advantages of the proposed method. The robustness of CATE estimation with respect to estimated nuisance functions is central to recent well-established works on CATE estimation that make use of semiparametric estimation theory, e.g., by Chernozhukov et al. (2018), Foster and Syrgkanis (2019), Nie and Wager (2021), Kennedy (2023). One of the key results is that CATE estimators with Neyman orthogonal loss functions (e.g., DR-learner, R-learner) are robust with respect to estimation errors of nuisance parameters (response surfaces, propensity score) in the sense of a fast guaranteed convergence rate. It is not clear to me that the proposed adversarial end-to-end approach improves on that. The proposed method already uses the Neyman orthogonal loss of the DR learner which makes the adversarial approach of optimizing for a pessimistic propensity score seem redundant.
- Despite being central to the topic of the paper, three of the four works mentioned above are not cited in the paper. Neither is the R-learner considered as a baseline.
- Furthermore, I do not understand why the proposed approach only performs adversarial learning w.r.t. the propensity score in combination with the doubly robust loss. Why not also for the response surfaces? Only accounting for the estimation errors in one nuisance parameter while ignoring the others seems arbitrary.
- A property of the DR loss is that only requires **either** the propensity score **or** the response surfaces to be estimated correctly to achieve a fast convergence rate (Kennedy, 2023). Again, this would make the proposed approach redundant if the response surface estimators converge sufficiently fast.
- While the method performs well in the experiments, the datasets seem to favor methods that focus on response function estimation rather than propensity score. PW-Net has a huge variance and the estimation error seems to grow with sample size for some reason. I suspect that there might be possible overlap violations in the simulated data. I could imagine that the proposed method offers some advantages in dealing with overlap violations, which might lead to an alternative way to frame the paper. However, this would require additional intuition and experiments.
- In summary, I think the problem of CATE estimation (or more generally, statistical estimation with nuisance parameters) is already quite well understood regarding the robustness to nuisance errors. I am not convinced that the proposed approach adds much benefit to the existing state-of-the-art.

Minor points

- The introduction puts a lot of emphasis on the CATE literature on representation learning. Personally, I do not think this literature stream is very relevant to the paper as it does not focus on representation learning, but CATE estimation as a statistical estimation problem with nuisance components. The same holds for the related work.
- The related work on CATE could be expanded.
- There are existing works on ATE estimation that estimate nuisance functions and ATE in an end-to-end manner (Shi et al. 2019, Frauen et al. 2023) which should be mentioned in the related work. However, these works perform end-to-end estimation to "target" the model parameters to fulfill estimation equations from semiparametric efficiency theory.
- In the literature, $\mu$ is usually used for the response functions and $\pi$ for the propensity score
- The consistency assumption is missing in Sec. 2

--- Post rebuttal response ---
I thank the authors for their detailed response, and in particular for adding the missing baselines. I raised my score. However, I am still not fully convinced that the proposed method improves on state-of-the-art approaches in "standard settings". While the authors argue that their method would outperform in high-dimensional settings (p > N), neither experiments nor theory are provided to support that claim. Furthermore, if I am not mistaken semiparametric state-of-the-art methods are also able to handle such settings.

On the other hand, I do believe that the proposed approach (performing adversarial learning specifically with respect to the propensity score) may have advantages in settings with low overlap when methods that divide by the propensity score (DR/ IPW) become unstable. Thus I see potential to frame the paper in that direction, and I would like to see a revised version at another venue. I believe it would be crucial to add experimental results that specifically show the benefit of the proposed method in settings with low overlap. E.g., plotting CATE estimation performance over the level of overlap. Furthermore, it would be important to compare with existing baselines that address overlap violations in such an experiment, e.g., R learner, or simply removing data points with low/large propensity scores. I hope that my comments are helpful for revising the current version of the paper.

**Questions:**

-Was data-splitting/ cross-fitting performed for the proposed method and the DR learner?

---

> ### Author Response · Authors · 2023-11-15
>
> Dear reviewer S5uy, we appreciate your detailed feedback. We understand the concerns and hope to address them below.
>
> ## Q1. The proposed method already uses the Neyman orthogonal loss of the DR learner which makes the adversarial approach of optimizing for a pessimistic propensity score seem redundant.
>
> Thank you for raising this quite important point of discussion.
> Since the current manuscript mainly discussed methods with deep models, there was only a limited description of parametric statistical methods in the introduction as follows.
>
> > Various countermeasures have been tried to alleviate this problem, such as doubly robust (DR) methods (Kang et al., 2007; Kennedy, 2020; Dudík et al., 2014), which are clever combinations of outcome prediction models and only weighting its residuals using estimated propensity. Nevertheless, the IPW-based approach’s limitation is the instability of the two-step procedure in non-asymptotic situations (Athey et al., 2018).
>
> As implied above, we believe that these (semi-)parametric analyses based on asymptotics assuming $p < N$ (at least w.r.t. the target $\tau$) have much room for improvement when applied to a complex (deep) model.
> The larger the hypothesis space, the more likely a hypothesis with an optimistic loss value will exist within it (and such a hypothesis is to be selected).
> In addition to the large hypothesis space, the extreme weights (even if accurate) exacerbate the effective sample size (Tanimoto et al. 2022, reference in the manuscript), which is usually ignored in the convergence rate analyses as a constant coefficient.
> Therefore we need a theory and a method that take care not only of the inaccuracy of the nuisance propensity model but also the estimation variance due to extreme weights.
>
> A series of DNN-based CATE estimation methods including (Shalit et al., 2017) avoid weighting, but this too has been pointed out to be theoretically limited (Johansson et al., 2019; Zhao et al., 2019; Wu et al., 2020).
> Some DNN-based methods such as (Shi et al. 2019; Hassanpour and Greiner, 2019) adopt simultaneous training of the model and weights; however, they adopt naive simultaneous training of them and lack a principle of their joint objectives.
>
> We would like to expand the description of the manuscript in this comparison.
>
> ## Q2. Despite being central to the topic of the paper, three of the four works mentioned above are not cited in the paper. Neither is the R-learner considered as a baseline.
>
> I would like to refer to the references you mentioned and clarify clarify along with our answer to Q1.
>
> We added the R-Learner as a baseline. See the overall response.
> R-Learner (RNet) exhibited similar trends with the DR-Learner.
> Our NuNet outperformed RNet in most settings.
>
> ## Q3. Why not also (adversarial) for the response surfaces? (S5uy)
>
> Estimation error is not the only reason for introducing the adversarial perturbation to the nuisance.
> Basically, the adversarial formulation would involve optimizing the concave objective function (maximizing the MSE), which reduces the stability of the training.
> Nevertheless, we believe that there is great merit in the adversarial formulation for the propensity, for the reasons stated in the above Q1 section.
> On the other hand, there is no such reason for the surfaces.
>
> In addition, the true effect and surfaces have a linear relation $\tau^*(x) = \hat f^*_1(x) - \hat f^*_0(x),$ which means that if the adversary adds perturbations $\Delta f_1$ and $\Delta f_0$, the learner can adjust as $\tau \leftarrow \tau + \Delta f_1 - \Delta f_0$ (ignoring the error correction terms), and the advantage for the adversary vanishes.
>
> We would like to clarify this in the manuscript.
>
> ## Q4. A property of the DR loss is that only requires **either** the propensity score **or** the response surfaces to be estimated correctly to achieve a fast convergence rate (Kennedy, 2023). Again, this would make the proposed approach redundant if the response surface estimators converge sufficiently fast.
>
> Actually, deep models are highly expressive and converge even with modeling surface learning alone.
> In order to speed up the convergence, models with flexible induction bias for the response surface are being explored.
> SNet is one such consideration.
> In practice, however, DRNet and NuNet with weighting often outperform SNet even without such a sophisticated model architecture (Figs. 2 and 3 in the manuscript).
> Therefore, it is important to have a framework that can utilize the expressive power of DNNs with weighting.

---

> > ### Author Response · Authors · 2023-11-15
> >
> > ## Q5. I suspect that there might be possible overlap violations in the simulated data. I could imagine that the proposed method offers some advantages in dealing with overlap violations
> >
> > Thank you for presenting a suggestive hypothesis for overlap violation.
> > This seems to have substantially in common with our hypothesis.
> > In a literal sense, our data generation process satisfies the overlap assumption (Appendix C1).
> > On the other hand, it may be very close to that (as is the case with real data), and the true density ratio (and thus the true weight) is to be very large.
> > In this situation, as noted in the Q1 section, accurate weighting would rather result in a large estimation variance.
> > This explanation is also consistent with the fact that methods using cross-fitting (presented in the overall response) have struggled.
> >
> > Again, we so much appreciate the very thorough discussion.

---

> ### Author Response · Authors · 2023-11-21
>
> We hope this message finds you well. We are writing to follow up on the rebuttal we submitted. We are keen to know if our rebuttal has adequately addressed your concerns.
>
> If our response has been satisfactory, we would greatly appreciate it if you could update the review status accordingly. If there are any further questions or additional clarification needed, please feel free to let us know. We welcome any additional feedback that can help improve our work.

---

### Official Review · Reviewer_DQuR · 2023-11-24

**Soundness:** 3 good
**Presentation:** 2 fair
**Contribution:** 2 fair
**Rating:** 5
**Confidence:** 3

**Summary:**

This paper proposes a CATE estimation method using doubly robust estimators and machine learning.

Here the outcome and propensity models are learned with ML in a "targeted" way to minimize the MSE of the CATE estimator, rather than being fit separately and plugged in. They then derive and bound/regularize additional loss terms that account for the contribution of nuisance mis-estimation to estimator bias.

**Strengths:**

- It's important to bridge the gap between ML and traditional yet challenging estimation problems such as CATE from observational data
- Theory looks good/correct
- Go beyond deriving a standard DR estimator and characterization additional issues with nuisance mis-estimation and what to do about it
- Experiments are pretty complete.

**Weaknesses:**

There seems to be substantial discussion of related work missing.

In particular, there is a lot of existing work that also directly estimates the nuisance functions with ML, and even does so in a regularized way to directly target the estimand.

Some examples include:
- Adapting Neural Networks for the Estimation of Treatment Effects, https://arxiv.org/abs/1906.02120
- RieszNet and ForestRiesz: Automatic Debiased Machine Learning with Neural Nets and Random Forests https://arxiv.org/abs/2110.03031

(EDIT: I saw that the Shi work was added in the revision. See "Questions")

More generally, as mentioned in questions below, I think there are some clarity issues, such as incomplete sentences, that make the work hard to understand. Some of those unclear sentences appear exactly when there is an important differentiation about related work to be made.

I would be willing to raise my score if the other reviewers believe that the related work has been clarified in the revision, and if the other reviewers and AC believe that unclear sentences/discussion points such as those above could be clarified easily.

**Questions:**

1)

(EDIT after revision) I saw that the updated draft includes the passage: "Joint optimization approaches have also been proposed for ATE estimation (Shi et al., 2019), though it may lead to cheating by less weighting to noisy regions especially under noise heterogeneity"

I am not sure what "cheating" means and what "less weighting" means?

2)

In the baseline section, in passing, the authors mention "DeR-CFR" (Wu et al., 2022) as another method that optimizes weights and outcome model simultaneously, but do not clarify why it is not compared against.

3)

"Although f0 and f1 are also nuisance parameters, the uncertainty of them do not differ among the target parameter space, thus we need not take care"

This sentence is very hard to understand; it is not complete and certain phrases are not defined "uncertainty does not differ" and "not taking care". Which uncertainty?

---

> ### Author Response · Authors · 2023-11-26
>
> ## W1. related work missing
>
> > * (Shi et al., 2019) Adapting Neural Networks for the Estimation of Treatment Effects, https://arxiv.org/abs/1906.02120
> > * (Chernozhukov et al., 2022) RieszNet and ForestRiesz: Automatic Debiased Machine Learning with Neural Nets and Random Forests https://arxiv.org/abs/2110.03031
>
> We have added (Shi et al., 2019) in our revision as you mentioned.
> The other one (Chernozhukov et al., 2022) is a similar approach based on Neyman orthogonality and (cooperative) joint optimization.
> These approaches share the same difficulties when considering their application to our setting.
>
> * Neyman orthogonality itself only guarantees the first-order robustness of nuisance estimation error. Higher-order also matters when the target model and nuisance are complex such as DNNs for CATE estimation rather than the ATE estimation (both of the above methods are for the ATE estimation).
> * A naive (cooperative, i.e. non-adversarial) joint objective is not justified just because the plug-in (two-step) estimation is justified. See the response to Q1.
>
> ## W2.  I think there are some clarity issues
>
> We would like to proofread the revised sentences that you mentioned in the camera-ready version (they cannot be revised immediately since the comments were received after the end of the discussion phase).
> See the responses to the questions below.
>
> ## Q1. I am not sure what "cheating" means and what "less weighting" means?
>
> In our revised manuscript, we wrote
>
> > Joint optimization approaches have also been proposed for ATE estimation (Shi et al., 2019), though it may lead to cheating by less weighting to noisy regions especially under noise heterogeneity.
>
> Naive joint optimization risks unintended weighting, which is the reason why we needed adversarial formulation.
> The weights are intended to adjust the distribution imbalance, as you know.
> However, another motive is introduced unintendedly when jointly optimized.
>
> Consider a simple objective function of the form $\min_{\mu, \theta} \sum_n w^{(n)}_{\mu} \ell^{(n)}$ with weights $w$.
>
> A trivial solution would be the $\mu$ that realize $w_\mu^{(n)} \to 0$, which of course can be prevented by normalization $\sum w^{(n)}_{\mu}=1$.
>
> On the other hand, when the noise levels differ across regions (the conditional variance $\mathbb{V}[\ell | x]$ is heterogeneous among $x$), the weights $w$ would be skewed toward easy instances (such $x$ that have small $\mathbb{V}[\ell | x]$) and hard instances are to be ignored.
> We call this "cheating."
>
> Such biased weights result in a biased loss surface in the target hypothesis space. A hypothesis with optimistic-side error (i.e., the loss is evaluated lower than the true value due to weights) tends to be selected in the training especially when the hypothesis space is large (e.g., complex CATE models) since the existence of such a highly optimistically evaluated hypothesis would be more probable.
>
> As shown in the additional experimental results (see below), naive joint optimization with reweighting makes the training quite unstable.
>
> ## Q2. "DeR-CFR" (Wu et al., 2022) as another method that optimizes weights and outcome model simultaneously, but do not clarify why it is not compared against.
>
> DR-CFR and DeR-CFR are similar approaches and SNet3 (Curth et al., 2021) summarizes them.
> SNet3 uses the same architecture with DR-CFR and DeR-CFR with a regularization that induces orthogonal representations proposed in DeR-CFR but omits reweighting.
> This is because, in our understanding, naive joint optimization with reweighting causes instability as we discussed above.
> We conducted an additional experiment comparing SNet3 with reweighting.
> Although we used the same hyperparameters as SNet3 at this moment and the regularization strength to avoid extreme weights can be further tunable, the results illustrate the instability of this naive approach.

---

> ### Author Response · Authors · 2023-11-26
>
> ## Q3. it is not complete and certain phrases are not defined "uncertainty does not differ" and "not taking care". Which uncertainty?
>
> In our revised manuscript, we wrote
>
> > Although $f_0$ and $f_1$ are also nuisance parameters, the uncertainty of them do not differ among the target parameter space, thus we need not take care.
> > Excluding the error correction term in (8), a disturbance ($\Delta f_0, \Delta f_1$) can be absorbed by translation $\tau \leftarrow \tau + \Delta f_1 - \Delta f_0$.
>
> The uncertainty we meant was the one that is induced to the target loss $L$ by $f_0,f_1$.
> The pessimistic (worst-case) loss surface leads to a different solution from the original (non-pessimistic) loss only when the uncertainty (confidence interval in $L$) differs among the hypotheses (if not, it would be meaningless as $\arg \min_h \max_{c \in [-1,1]} L(h) + c = \arg \min_h L(h)$).
> Similarly, the essential part of the loss would be like following.
> Consider a scalar case and let $t = f_1 - f_0$ be the target. Then,
> $$\arg \min_\tau \max_{\Delta t \in [-1, 1]} (t + \Delta t - \tau)^2 = \arg \min_\tau (t - \tau)^2$$
> due to the convexity and isotropy of $(t - \tau)^2$ on $\tau$.
> It does not exactly reproduce the situation in our setting, but we do not consider disturbing $f_0$ and $f_1$ meaningful for essentially the same reason.
>
> The other reason is that unlike propensity scores, where it makes sense (in the sense of equation (3)) to avoid extreme weights at the expense of accuracy for $p(t|x)$, there is no positive advantage to moving them significantly from the pre-trained $f_1, f_0$.
> Please also refer to the response for Q3 of Reviewer S5uy.
>
> ## Additional results
>
> We additionally compared SNet3 with reweighting with the same hyperparameter setting as SNet3.
>
> * Additive noise setting
> | Method     | N=500             | 1000              | 2000              | 5000              | 10000             | 20000             |
> |------------|-------------------|-------------------|-------------------|-------------------|-------------------|-------------------|
> |SNet3	| 13.10  ±0.65 	| 7.73 ± 0.34  	|   3.85 ± 0.11 	|   1.54 ± 0.05 	|   0.99 ± 0.02 	|   0.62 ± 0.01  |
> |SNet3 w/ reweighting	| 61.47 ± 2.78  | 63.13 ± 2.32	| 63.79 ± 2.27	| 69.28 ± 1.30	| 72.20 ± 1.82	| 74.67 ± 1.89 |
> |DRNet       |16.56 ± 0.75       |11.58 ± 0.66       | 3.91 ± 0.14       | 1.45 ± 0.04       | 1.14 ± 0.11       | 0.66 ± 0.03       |
> |NuNet (proposed)      |15.78 ± 0.69       |11.43 ± 0.48       | 4.02 ± 0.09       | 1.52 ± 0.07       | 0.86 ± 0.01       | 0.54 ± 0.01       |
>
> * Multiplicative noise setting
> | Method     | N=500             | 1000              | 2000              | 5000              | 10000             | 20000             |
> |------------|-------------------|-------------------|-------------------|-------------------|-------------------|-------------------|
> |SNet3	|17.83 ± 0.94	|  15.44 ± 0.65	|  11.12 ± 0.36	|   5.71 ± 0.25	|   3.61 ± 0.14	|   2.46 ± 0.09 |
> |SNet3 w/ reweighting	| 60.87 ± 2.58	| 61.92 ± 2.04	| 64.04 ± 1.93 	| 67.63 ± 0.80	| 70.63 ± 1.73 |	74.21 ± 1.60 |
> |DRNet       |19.96 ± 1.01       |15.34 ± 0.75       | 9.93 ± 0.40       | 4.80 ± 0.21       | 3.20 ± 0.24       | 1.83 ± 0.10       |
> |NuNet (proposed)     |19.96 ± 1.20       |15.54 ± 0.57       | 9.83 ± 0.45       | 4.67 ± 0.32       | 2.44 ± 0.09       | 1.50 ± 0.06       |
>
> We hope our response clears your concerns.

---

### Author Response · Authors · 2023-11-15
**Overall response**

Dear Reviewers,

We are grateful for the efforts and taking the time to our paper and for the constructive feedback.

## Positive feedback

Reviewers agree that:
* The problem is *important* (S5uy) and the motivation is *clear* (KLj8)
* Our work is *well written* (S5uy)
* The proposed method *performs well* empirically (S5uy) and *robust* to many settings (KLj8).

## Concerns and responses (summarized)

On the other hand, the reviewers expressed the following concerns.

1. Orthogonal learning methods (e.g., DR-Learner and R-Learner) are already robust to the misestimation of the nuisance propensity.

We believe that the convergence rate guarantee in (semi-)parametric inference is sufficient where sample size dominates relative to the complexity of the hypothesis space, but has much room for improvement when applied to complex models such as DNNs.
In addition to the normal estimation error, the propensity *should* be biased to avoid extreme weights, since the DNN-based propensity models *can* express extreme propensity values.
Our method is the first theory and method, to our knowledge, that can simultaneously take care of the estimation variance due to extreme weights and the bias due to incorrect propensity scores to avoid the extreme weights.

2. Some baselines (R-learner and cross-fitting) are missing (S5uy)

We added results. See below.

3. Why not also (adversarial) for the response surfaces? (S5uy)

Because there is no point in disturbing the response surfaces ($f$s).
The effect model is trained so that $\tau(x) \simeq \hat f_1(x) - \hat f_0(x) $ (+ error correction terms) as in Eq. (8) (i.e., a linear relation between $\tau$ and $f$s, for the main part).
If adversary add $\Delta f_1$ and $\Delta f_0$, the learner can adjust as $\tau \leftarrow \tau + \Delta f_1 - \Delta f_0$.
It just makes the training unstable (due to $f$'s concavity to the MSE).
We would like to clarify this.

4. A baseline method (SNet) is better in some settings (KLj8)

SNet's sophisticated architecture seems to exploit intrinsic structures inside the data generation process.
SNet has shared representation layers among the propensity model and the two potential outcome predictors.
Exploring architectures with better inductive bias is an important direction that is separate from and orthogonal to our study, and combining them is promising future work.

## Additional results

Motivated by reviewer S5uy's comments, we added several baselines (DR-Learner with cross-fitting, R-Learner with and without cross-fitting) to our experiment on synthetic datasets.
The results show that R-Learner (RNet) performs as well as DRNet, and cross-fitting makes them worse.
Without cross-fitting, the estimated propensity score of $(a^{(n)}, x^{(n)})$, which is actually very low probability, $\hat \mu(a^{(n)}|x^{(n)})$, is estimated larger than it actually is, resulting in a milder weighting.
This demonstrates the advantage of avoiding extremely small values rather than pursuing accuracy in propensity scores.

* Additive noise setting
| Method     | N=500             | 1000              | 2000              | 5000              | 10000             | 20000             |
|------------|-------------------|-------------------|-------------------|-------------------|-------------------|-------------------|
| RNet	    | 18.47 ± 3.25	    | 13.65 ± 1.14	    | 6.88 ± 0.40	    | 2.31 ± 0.07	    | 1.41 ± 0.07	    | 1.02 ± 0.04       |
| RNet w/ cross-fit	| 15.28 ± 0.90	    | 11.79 ± 0.50	    | 6.46 ± 0.27	    | 2.63 ± 0.08	    | 1.47 ± 0.04	    | 0.92 ± 0.03       |
|DRNet       |16.56 ± 0.75       |11.58 ± 0.66       | 3.91 ± 0.14       | 1.45 ± 0.04       | 1.14 ± 0.11       | 0.66 ± 0.03       |
|DRNet w/ cross-fit	|162.27 ± 75.79	    |18.83 ± 1.55	    | 9.07 ± 0.53       | 6.51 ± 1.38	    | 5.90 ± 1.19	    | 3.94 ± 1.41       |
|NuNet (proposed)      |15.78 ± 0.69       |11.43 ± 0.48       | 4.02 ± 0.09       | 1.52 ± 0.07       | 0.86 ± 0.01       | 0.54 ± 0.01       |

* Multiplicative noise setting
| Method     | N=500             | 1000              | 2000              | 5000              | 10000             | 20000             |
|------------|-------------------|-------------------|-------------------|-------------------|-------------------|-------------------|
| RNet	    | 32.26 ± 6.86	    | 11.87 ± 0.30	    | 9.18 ± 0.45	    | 5.12 ± 0.16	    | 2.98 ± 0.09	    | 1.91 ± 0.06       |
| RNet w/ cross-fit	| 18.74 ± 1.34	    | 13.98 ± 0.62	    | 10.64 ± 0.49	    | 5.81 ± 0.16	    | 3.45 ± 0.08	    | 1.99 ± 0.06       |
|DRNet       |19.96 ± 1.01       |15.34 ± 0.75       | 9.93 ± 0.40       | 4.80 ± 0.21       | 3.20 ± 0.24       | 1.83 ± 0.10       |
| DRNet w/ cross-fit	|179.60 ± 116.17	|21.65 ± 1.95	    | 12.80 ± 0.70	    | 10.42 ± 0.75	    | 9.68 ± 1.49	    | 7.14 ± 1.32       |
|NuNet (proposed)     |19.96 ± 1.20       |15.54 ± 0.57       | 9.83 ± 0.45       | 4.67 ± 0.32       | 2.44 ± 0.09       | 1.50 ± 0.06       |

---

### Meta-Review · Area_Chair_F4Bg · 2023-12-14

**Metareview:**

This paper tries to push weighting schemes to improve them relative to representation balancing approaches for causal inference. While the question was seen as interesting to reviewers and CATE estimation an important question, the main issue was how this work fits in with the relatively large literature on CATE estimation that considers nuisance functions. Both of the negative reviewers updated their reviews after the author response and updated pdf, one raised their score. Yet after the update, there were still questions about where the proposed approach would have value with respect to existing work. See the post rebuttal from (reviewer S5uy).

I asked the positive reviewer if they felt strongly. They said the agree with the two more negative reviewers major concerns, so did not care to adhere to the initial positive evaluation.

**Justification For Why Not Higher Score:**

The paper is not clear about it where it would have advantages to related work and more broadly how it sits within the area.

**Justification For Why Not Lower Score:**

NA

---

### Decision · Program_Chairs · 2024-01-16

Reject